# Construction and Validation of a Nomogram Clinical Prediction Model for Predicting Osteoporosis in an Asymptomatic Elderly Population in Beijing

**DOI:** 10.3390/jcm12041292

**Published:** 2023-02-06

**Authors:** Jialin Wang, Chao Kong, Fumin Pan, Shibao Lu

**Affiliations:** Department of Orthopedics, Xuanwu Hospital, Capital Medical University, No. 45 Changchun Street, Beijing 100000, China

**Keywords:** osteoporosis, clinical prediction model, nomogram, asymptomatic elderly, early diagnosis, screening

## Abstract

Background: Based on the high prevalence and occult-onset of osteoporosis, the development of novel early screening tools was imminent. Therefore, this study attempted to construct a nomogram clinical prediction model for predicting osteoporosis. Methods: Asymptomatic elderly residents in the training (*n* = 438) and validation groups (*n* = 146) were recruited. BMD examinations were performed and clinical data were collected for the participants. Logistic regression analyses were performed. A logistic nomogram clinical prediction model and an online dynamic nomogram clinical prediction model were constructed. The nomogram model was validated by means of ROC curves, calibration curves, DCA curves, and clinical impact curves. Results: The nomogram clinical prediction model constructed based on gender, education level, and body weight was well generalized and had moderate predictive value (AUC > 0.7), better calibration, and better clinical benefit. An online dynamic nomogram was constructed. Conclusions: The nomogram clinical prediction model was easy to generalize, and could help family physicians and primary community healthcare institutions to better screen for osteoporosis in the general elderly population and achieve early detection and diagnosis of the disease.

## 1. Introduction

Osteoporosis (OP), as a systemic skeletal disease, is increasingly being focused on by clinical practitioners in both orthopedics and endocrinology [1]. As the population ages and the average life expectancy of people increases, the incidence of osteoporosis is on the rise year by year, and this is even more pronounced in China, because China is rapidly entering an aging society [2,3]. With the development of osteoporosis, the malignant prognosis of decreased quality of life, reduced mobility, the occurrence of fragility fractures, and even death has become a topic that older adults and clinical researchers have to face, which also brings a huge social and economic burden [4,5]. However, osteoporosis is insidious at the onset, early, and even late stages of the disease, and a large proportion of patients are only diagnosed with osteoporosis when a fragility fracture occurs [6]. Therefore, early screening, prediction, and diagnosis of osteoporosis are essential. The development of scientific, rational, and simple early clinical screening, prediction, and diagnosis tools is urgently needed.

The diagnosis of osteoporosis and the determination of bone mineral density (BMD) are varied and include ultrasound, dual X-ray absorptiometry (DXA), and quantitative computed tomography (QCT) [7,8,9]. Among them, BMD measurement by DXA and calculation of a T-score is currently the most common and accurate test for the diagnosis of osteoporosis, and is recognized by researchers as the gold standard [10,11]. According to the recommendation of the World Health Organization (WTO) in 1994, a T-score less than or equal to −2.5 standard deviations (T ≤ −2.5 SD) is considered osteoporosis for postmenopausal elderly women and elderly men over 50 years of age [12]. However, DXA examinations are essentially X-rays and have been shown to have effects on multiple systems and multiple diseases throughout the body, which is their inevitable drawback [13,14]. On the other hand, BMD screening requires a certain cost and expense, which poses a great difficulty for its widespread dissemination and use for broad-spectrum screening of the masses, especially in China, which is still a developing country with uneven regional economic development [15]. Therefore, is it possible to perform a simple prediction and screening in the early stage, and then recommend DXA to only a portion of the screened population in order to increase the detection rate while reducing the side effects and costs associated with DXA?

Clinical predictive models are now widely used in the clinic for the diagnosis of a particular disease or the prediction and evaluation of a clinical outcome [16]. Among them, logistic regression, cox regression, nomogram, lasso regression, and machine learning models have been increasingly used in recent years [17,18,19,20,21]. Of these, nomogram prediction models have been applied in the diagnosis and prognosis of various diseases including prostate cancer, colorectal cancer, gastric cancer, osteoporosis, and even COVID-19 pneumonia, and the predictive efficacy of these nomogram prediction models has also been proven [22,23,24,25,26]. For example, a study by Liu et al. published in *Lancet Digital Health* showed that nomogram prediction models based on predictors such as age, Glasgow Coma Score (GCS), and respiratory rate were able to predict the risk of noninvasive respiratory failure in patients with COVID-19 [27]. In addition, it has been shown that nomogram prediction models based on genetic and epidemiological risk factors can provide some individual predictive performance for leprosy in southwest China [28]. However, there are relatively few studies on nomogram prediction models for osteoporosis, and there are no studies on nomogram prediction models related to risk prediction of osteoporosis in the asymptomatic elderly in China.

Therefore, the development of tools for the early diagnosis of occult-onset osteoporosis is urgent, but there is a lack of nomogram prediction models for predicting osteoporosis. This study aims to construct and validate a nomogram prediction model for predicting osteoporosis in asymptomatic elderly people in Beijing by collecting BMD and health information from asymptomatic elderly people in order to achieve early screening of osteoporosis in the general community and a certain degree of clinical translation.

## 2. Materials and Methods

### 2.1. Study Design and Population

This study was conducted as a cross-sectional study, and elderly residents from several residential communities in Beijing were invited to participate. The inclusion criteria for this study were (1) elderly males or postmenopausal elderly females over 50 years of age; (2) able to undergo bone mineral density screening; (3) more than 5 years of residential history in the Beijing area; (4) able to complete the questionnaire and provide basic health information; and (5) voluntarily participated in this study and signed an informed consent form. The exclusion criteria for this study were (1) history of lumbar spine surgery or hip surgery; (2) significant discomfort such as low back pain with a VAS score > 3; (3) those with mental illness/limited mobility or communication impairment; and (4) history of malignancy.

A total of 1250 elderly participants were recruited for this study and, after inclusion and exclusion criteria, a total of 584 residents eventually participated. Among them, 438 participants from June 2021 to December 2021 were included in the training group, and 146 participants from January 2022 to May 2022 were included in the validation group.

This study was approved by the Ethics Committee on Biomedical Research, West China Hospital of Sichuan University. All patients signed a written informed consent form.

### 2.2. Information on Clinical Factors

Subjects’ various common osteoporosis-related clinical factors were collected by questionnaire and standardized measurements. These clinical factors included age, gender, manual laborer or not, education level, height, weight, waistline, history of smoking, and history of drinking. Educational level was categorized as “Junior high school”, “High school”, and “Undergraduate”. History of drinking was defined as drinking liquor at least once a week, more than 50 mL each time, lasting more than one year, still drinking, or not quitting alcohol for more than 3 years. The three clinical factors of height, weight, and waistline were measured by experienced staff following standard procedures. The subject’s height was measured from the bottom of the foot (without shoes) to the top of the head using a straightedge (0.1 cm accuracy). The weight of the subjects (wearing light clothing indoors) was measured using an electronic scale (accuracy 0.1 kg). The waistline (waist circumference past the navel) of the subjects was measured using a tape measure (accuracy 0.1 kg). The relevant physical measurements were repeated twice and the results were averaged.

### 2.3. DXA Examination and Diagnostic Criteria for Osteoporosis

Bone density at the lumbar spine area and at the right and left hips (proximal femur) was measured by dual-energy X-ray absorptiometry (GE Lunar iDXA, GE Healthcare, Madison, WI, USA). A professional technician calibrated the machine according to the model provided by the manufacturer and performed routine quality control and quality assurance of the machine according to the manufacturer’s instructions.

The BMD (g/cm^2^) was assessed by a radiologist blinded to this study. According to the recommendation of the World Health Organization (WTO) in 1994, for postmenopausal elderly women and elderly men over 50 years of age, a T-score greater than or equal to −1.0 standard deviation (T ≥ −1.0 SD) is considered normal BMD; a T-score between −1.0 and −2.5 standard deviations (−1.0 SD > T > −2.5 SD) is considered osteopenia; a T-score less than or equal to −2.5 standard deviations (T 2.5 standard deviations (T ≤ −2.5 SD)) is considered osteoporosis, and the presence of osteopenia with one or more fractures is considered severe osteoporosis.

### 2.4. Construction and Validation of the Nomogram Prediction Model

The nomogram prediction model was constructed and validated based on the training and validation groups. Since the outcome variable (presence of osteoporosis) is a dichotomous variable, the logistic regression nomogram prediction model was selected according to the nomogram prediction model construction guidelines [29]. Based on the results of univariate and multivariate logistic regression analyses and certain clinical experience, specific clinical variables were selected as predictors for the nomogram prediction model. Regression coefficients, odds ratios (ORs), and 95% confidence intervals (CIs) were then calculated for each predictor. The regression coefficients of each predictor were used as weights to draw the nomogram. The dynamic nomogram was constructed and published online according to the online tool from shinyapps.

After that, the performance and accuracy of the nomogram prediction model were verified by a series of methods for the training group and the validation group, respectively. Calibration curves based on the Hosmer–Lemeshow goodness-of-fit test were constructed to verify the calibration of the nomogram prediction model. The receiver operating characteristic curve (ROC) was constructed, and the area under the curve (AUC) and concordance index (C-index) were calculated to validate the discrimination of the nomogram prediction model. The AUC values and C-index of the two cohorts (the training and validation groups) were calculated by 3-fold cross-validation, 5-fold cross-validation, 10-fold cross-validation, Jackknife validation, and Bootstrap validation. The Decision curve analysis (DCA curves) and clinical impact curves were constructed to validate the clinical benefit and clinical impact of the nomogram prediction model. The above process was mainly performed by R software (version 4.2.1) and the relevant codes were documented in Appendix A.

### 2.5. Statistical Analysis

The statistical analysis of this study was carried out mainly through R software (version 4.2.1, Boston, MA, USA) and SPSS software (version 26.0, Chicago, IL, USA). The continuous variables of this study were tested for normality by the Kolmogorov–Smirnov test. Normality continuous variables were presented as mean ± standard deviation and the statistical significance of the differences between two groups was analyzed by the Student’s *t*-test. Non-normal continuous variables were presented as median ± upper and lower quartiles, and the statistical significance of the differences between the two groups was analyzed by the Kruskal–Wallis H-test. Categorical variables were presented as frequencies and percentages, and the statistical significance of the differences between the two groups was analyzed using the chi-square test. The interaction between the variables was evaluated after their variable transformation. Continuous variables were transformed into dichotomous variables according to their means, and then analyzed for interaction effects. Univariate and multivariate logistic regression analyses were performed to explore the risk factors for osteoporosis. The receiver operating characteristic curve (ROC) and area under the curve (AUC) were used to assess the discrimination and predictive value of the nomogram prediction model, which was graded as (1) 0.5 < AUC ≤ 0.7, low predictive value, (2) 0.7 < AUC ≤ 0.9, moderate predictive value, and (3) 0.9 < AUC < 1, high predictive value. The grading of the concordance index (C-index) was equivalent to the AUC value. In this study, *p* < 0.05 was considered to be statistically significant.

## 3. Results

### 3.1. Participant and Clinical Characteristics

The main design and steps of this study were shown in Graphical Abstract. Baseline information on the clinical characteristics of the participants in the training and validation groups were presented in Table 1 and Appendix A. Appendix A shows the group comparisons between the training and validation groups. As shown in Appendix A, there were 438 participants in the training group, of whom 185 had osteoporosis and 253 had no osteoporosis, for an osteoporosis prevalence of 42.2%. The validation group consisted of 146 individuals, of whom 63 had osteoporosis and 83 had no osteoporosis, with an incidence of 43.1%. Meanwhile, there were no significant differences between the training group and the validation group in terms of clinical characteristics such as age, gender, manual laborer or not, education level, height, weight, waistline, and history of smoking and drinking. These ensure the validation effectiveness of the validation group on the nomogram prediction model constructed from the training group in the follow-up study.

Table 1 shows the intergroup comparisons between the various subgroups (non-osteoporotic and osteoporotic groups) of the training and validation groups. On the aspect of age, the results showed that the average age of the participants in both groups (training and validation groups) was 67.2 and 66.1 years, respectively, which was in line with the objective of this study for the elderly. The results also showed that the average age of the participants in the osteoporotic group was higher than in the non-osteoporotic group in both groups (training and validation groups), but this difference was not statistically significant (training group, *p* = 0.083; validation group, *p* = 0.240). For gender as a clinical characteristic, the percentage of osteoporosis was significantly higher in the female group in both the training and validation groups (*p* < 0.001). In terms of manual laborer or not, no statistically significant group differences were found. For education level, participants with “High school” and “Undergraduate” education levels had a lower probability of osteoporosis than those with “Junior high school” education level (training group, *p* < 0.001; validation group, *p* = 0.001). The results also suggested that the median height, median weight, and median waistline of the participants in the osteoporotic group were significantly lower than those in the non-osteoporotic group in both groups (training and validation groups) (*p* < 0.001). For the history of smoking and drinking, the results suggested that, to some extent, the participants in the non-osteoporotic group had higher rates of smoking and alcohol consumption, which was considered as possibly influenced by gender factors.

In conclusion, intergroup comparisons of these subgroups were somewhat suggestive of an association between individual clinical characteristics and osteoporosis, but the identification of specific risk factors still requires further study.

### 3.2. Univariate and Multivariate Logistic Regression Analyeis of Osteoporosis

After exploring the differences in baseline information between the different subgroups, an analysis of interactions was performed for each clinical characteristic to exclude internal interference between clinical characteristics. The analysis revealed no significant interactions between most clinical characteristics (Appendix A). Afterwards, risk factors for osteoporosis were explored by univariate and multivariate logistic regression analyses for the training and validation groups, respectively. As shown in Figure 1, in the training group, the results of univariate analysis for women (with men as the reference) were (OR = 3.55, 95% CI = 2.18–5.79, *p* < 0.001), while the results of multivariate analysis were (OR = 2.92, 95% CI = 1.21–7.06, *p* = 0.017). The results of univariate analysis of education level at High school level (with Junior high school as the reference) were (OR = 0.31, 95% CI = 0.20–0.50, *p* < 0.001), while the results of multivariate analysis were (OR = 0.40, 95% CI = 0.24–0.67, *p* = 0.017). The results of univariate analysis of college level education (with Junior high school as the reference) were (OR = 0.25, 95% CI = 0.15–0.43, *p* < 0.001), while the results of multivariate analysis were (OR = 0.47, 95% CI = 0.25–0.87, *p* = 0.017). The results of univariate analysis for higher weight were (OR = 0.90, 95% CI = 0.88–0.92, *p* < 0.001), while the results of multivariate analysis were (OR = 0.91, 95% CI = 0.88–0.94, *p* ≤ 0.001). The results of univariate and multivariate logistic regression analyses for the validation group are presented in Figure 2. The results in Figure 2 also illuminated and validated the results in Figure 1. However, the results for the High school level of education in Figure 2 did not match the results in Figure 1, which was considered to be the reason for the lower sample size in the validation group compared to the training group. Finally, Appendix A are the presentation forms of the tables of the results of univariate and multivariate logistic regression analyses.

Therefore, based on the above univariate and multivariate logistic regression analyses results, it could be hypothesized that, among women, lower education level and lower weight were independent risk factors for osteoporosis.

### 3.3. Construction and Validation of the Nomogram Clinical Prediction Model for Osteoporosis

Based on the results of univariate and multivariate analyses in the previous paper and combined with clinical experience, gender, education level, and weight were selected as predictors of the nomogram clinical prediction model for osteoporosis. The constructed nomogram clinical prediction model of osteoporosis is shown in Figure 3. By visualizing this figure, it was possible to make accurate and individualized clinical predictions for clinical participants by simple operations such as marking points, making vertical lines and adding, and achieving better clinical translation. The nomogram in Figure 3 also shows an example based on specific information of a clinical participant. In addition, Figure 3 shows to some extent the distribution of the three clinical predictors mentioned above. These above can be seen specifically in the figure legend in Figure 3. In addition, an online dynamic nomogram prediction tool for osteoporosis was developed in this study (Figure 4). After accessing this link (https://shibaolu.shinyapps.io/DynamicNomogram/ accessed on 22 December 2022) on any mobile device, mobile, convenient, and visual prediction could be achieved with a simple click, select, and swipe button, which was ideal for patient-physician communication and personalized clinical treatment plan decisions. These can be seen in the figure legend in Figure 4.

After constructing the general and online dynamic nomogram prediction model, this study also attempts to verify the prediction performance and scientific validity of the prediction model through a scientifically sound approach. Figure 5 shows the ROC curves and calibration curves for the training and validation groups. Figure 5A shows that for the training group, the prediction performance of the nomogram clinical prediction model was (AUC = 0.761, 95% CI = 0.717–0.805, cutoff value = 0.373, precision = 0.601, sensitivity = 0.784). Figure 5B shows that, for the validation group, the prediction performance of this nomogram clinical prediction model was (AUC = 0.765, 95% CI = 0687–0.842, cut-off value = 0.355, precision = 0.578, sensitivity = 0.857). Thus, the results of both Figure 5A,B suggested that the nomogram clinical prediction model for osteoporosis developed in this study has moderate predictive value (AUC > 0.7). In contrast, Figure 5C,D shows that the Apparent and Bias-corrected lines, which represented the actual nomogram model, did not deviate much from the Ideal line which represented the ideal state for both the training and validation groups, suggesting that the nomogram clinical prediction model for osteoporosis developed in this study had a certain degree of calibration. Next, Figure 6 shows the DCA curves and clinical impact curves for the training and validation groups. Figure 6A shows that, for the training group, the nomogram clinical prediction model was able to provide clinical benefit to the clinical participants at the judgment threshold between 0 and 0.75. Figure 6B shows that, for the validation group, the nomogram clinical prediction model was able to provide clinical benefit to the clinical participants at the judgment threshold between 0 and 0.90. Figure 6C,D then shows that, for both the training and validation groups, to some extent, the nomogram clinical prediction model was able to provide good clinical impact for clinical participants.

In summary, this study constructed and validated a nomogram clinical prediction model for predicting osteoporosis. The clinical prediction model is not only simple, visual, easy to operate, easy to generalize, and easy to translate clinically, but also has been scientifically and rationally validated to demonstrate its appropriate discrimination, calibration, predictive accuracy, and clinical benefit.

## 4. Discussion

The insidious onset of osteoporosis makes early treatment of the disease difficult, while the invasive and costly nature of DXA makes broad-spectrum screening impossible in developing countries and regions [1,13]. Therefore, the development of simple and practical personalized early prediction tools is essential and clinically relevant. On the other hand, nomogram clinical prediction models offer a bright future for diagnostic and prognostic assessment of multiple diseases [27]. Therefore, in this study, DXA examinations and basic health information of asymptomatic residents in residential communities were collected and a clinical prediction model for osteoporosis was constructed and validated specifically for general asymptomatic residents, with the aim of early screening and prediction of osteoporosis at the family physician level or community primary hospital level to aid clinical decision making and personalized treatment.

First, in this study, bone mineral density examinations and basic health information were collected from a total of 584 participants in the training group (*n* = 438) and validation group (*n* = 146). The results in Appendix A suggested that there were no significant differences between the training and validation groups in terms of individual clinical characteristics, which was a prerequisite for adequate validation efficacy in the later validation group. Then, the results showed that the probability of osteoporosis in the training and validation groups were 42.2% and 43.1%, respectively, which was higher than the probability of osteoporosis in previous studies [30]. This study suggested that the reason for this is mainly due to age and gender differences in the different study groups. The average age of the population in this study was 66 years, which was higher than other studies [31]. Also, over 70% of the participants in this study were women, and, considering that most studies have shown a higher probability of osteoporosis in women, it was acceptable to show a higher probability of osteoporosis in this study [31]. In addition, older women tend to be more concerned about their personal health than older men, which may be one of the reasons why more older women were willing to participate in this study. Later, Table 1 also shows that participants in the osteoporosis group had lower education level, lower height, lower weight, and lower waistline, which was corroborated by several previous studies [32]. Subsequent interaction analysis in Appendix A was able to rule out some covariates and other issues. The results of univariate and multivariate logistic regression analyses in Figure 1 and Figure 2 and Appendix A suggested that female, lower education level, and lower weight were independent risk factors for osteoporosis. Among them, women were independent risk factors for osteoporosis, which has been confirmed in several national and regional studies [1]. In addition, higher levels of education tend to be accompanied by better economic status and greater concern for health, and it has also been shown that the prevalence of osteoporosis is higher in Chinese men with lower levels of education [33]. It is of interest that the participants in this study were mainly from Beijing, which has a high level of education compared to other regions. On the other hand, higher body weight implied a higher stress on the bone fiber structure and also tended to be accompanied by a better nutritional status. Some studies have shown an increased risk of fracture in several ministries after weight loss in postmenopausal older women [34]. The above results demonstrated some baseline information about the participant group of this study and also identified independent risk factors (female, lower education level, and lower weight) for osteoporosis for the asymptomatic elderly resident community.

Subsequently, the nomogram clinical prediction model for predicting osteoporosis was constructed and validated through a series of canonical methods [29]. First, considering the dichotomous variable of predicting the outcome as whether it was osteoporosis, the selection of the logistic nomogram model was mandatory. Second, the results in Appendix A showed no significant differences between the training and validation groups in terms of multiple clinical factors, which ensured the validation efficacy of the validation group and the generalizability of the nomogram model. After that, based on the results of univariate and multivariate logistic regression analyses, combined with clinical experience, it was decided to select three variables—gender, education level and weight—as predictors for the nomogram prediction model. According to Harrell’s guidelines, the number of predictors must be less than one-tenth of the overall sample size, which was met in the present study [29]. Prediction methods or models with a smaller number of predictors tended to have higher simplicity and ease of generalization, which was advantageous for diseases that require large-scale screening, such as osteoporosis. This could be verified in a variety of well-established early screening methods that have been developed previously. For example, the Osteoporosis Self-Screening Tool for Asians (OSTA) and the Osteoporosis Self-Screening Tool for Chinese (OSTC) require only two predictors—age and weight—which has allowed for their widespread dissemination [35,36]. However, these two prediction tools have their limitations in that they can only predict a range and do not give an accurate probability figure for osteoporosis in subjects. The nomogram developed in the present study was a good remedy for this shortcoming. As shown in Figure 3, the nomogram could allow the clinician to mark points, make vertical lines, and add them up to get an accurate probability of osteoporosis, which could obviously help with communication between the physician and the patient. In addition, family physicians and primary community healthcare providers can access the online dynamic nomogram (Figure 4) via the link (https://shibaolu.shinyapps.io/DynamicNomogram/ accessed on 22 December 2022) from any mobile device, which undoubtedly greatly increases the feasibility of screening for osteoporosis based on this nomogram clinical prediction model. Finally, a scientifically sound validation of the nomogram prediction model was essential. Figure 5 suggests that the nomogram prediction model had a moderate predictive value (AUC > 0.7) and good calibration. Figure 6 suggests that the nomogram prediction model provided a good clinical benefit to clinical participants. Therefore, the above results confirmed the rationality, scientific validity, and generalizability of the nomogram prediction model developed in this study for predicting osteoporosis in asymptomatic elderly people in the community. Through the common and online dynamic interactive nomogram prediction model constructed in this study, a quick and simple screening of asymptomatic elderly people in the community could be carried out in practice. If the subject’s “probability of osteoporosis” is high (0.74), it should be recommended that the subject undergoes DXA examination as much as possible, which can improve the detection rate of osteoporosis to a certain extent. If the subject is indeed in the early stage of osteoporosis—for example, the T-score of BMD in the later test is −2.6 SD— the subject could be given certain anti-osteoporosis treatment to achieve early detection of osteoporosis and early treatment, thereby reducing the rate of osteoporotic fracture and mortality. If the subject’s “probability of osteoporosis” is medium (0.51), and the subject has a high probability of developing osteoporosis—for example, the T-score of BMD in the subsequent test is −2.3 SD—the subject could be given certain treatment to prevent osteoporosis, thereby reducing the incidence of osteoporosis. The above discussion about the potential impact of this study on osteoporosis morbidity and mortality in practice demonstrates its potential clinical significance and value to public health.

In general, this study has certain advantages. First, the participants of this study were mainly asymptomatic elderly people recruited from the community rather than from hospital physical examination centers, which ensures the generalizability of the nomogram clinical prediction model developed in this study to a certain extent. Second, this study was divided into a training group and a validation group, which was necessary according to the relevant guidelines for the construction of the nomogram [29]. After that, the online dynamic nomogram developed was very practical and easy to generalize. Finally, the validation method of the nomogram clinical prediction model was adequate and reasonable. However, there were still some limitations in this study. The study participants were mainly from the Beijing area, which made the nomogram clinical prediction model potentially limited in its generalization to the south or other regions. On the other hand, there may be more potential risk factors to consider in other regions or populations because, if a more comprehensive risk factor is included, it will inevitably affect the validation indicators such as the discrimination and calibration of the nomogram model. This is also a great challenge to the generalizability of the nomogram clinical prediction model developed in this study. For example, in areas with unbalanced economic development, the economic level of the subjects should be considered. Other examples may include the subject’s exercise status, nutritional intake, and use of anti-osteoporosis drugs. Therefore, it is urgent to explore the potential risk factors of osteoporosis more comprehensively in subsequent studies using populations from other regions as validation groups in order to validate the nomogram clinical prediction model more reasonably. In addition, the research outcome of this study is the occurrence of osteoporosis but, to some extent, the three-year, five-year or ten-year incidence rate of osteoporosis can be assessed by exploring the results with a time span. It is also a very meaningful future research direction.

## 5. Conclusions

In conclusion, based on the BMD results and clinical data of the asymptomatic elderly population in the Beijing community, this study found that women, lower education level, and lower weight were independent risk factors for osteoporosis. The simple and easily replicable nomogram clinical prediction model constructed on this basis could help family physicians and primary community healthcare institutions to conduct reasonable screening for osteoporosis in residents, thus playing a role in early screening and early diagnosis of osteoporosis and achieving some clinical translation.

## Figures and Tables

**Figure 1 jcm-12-01292-f001:**
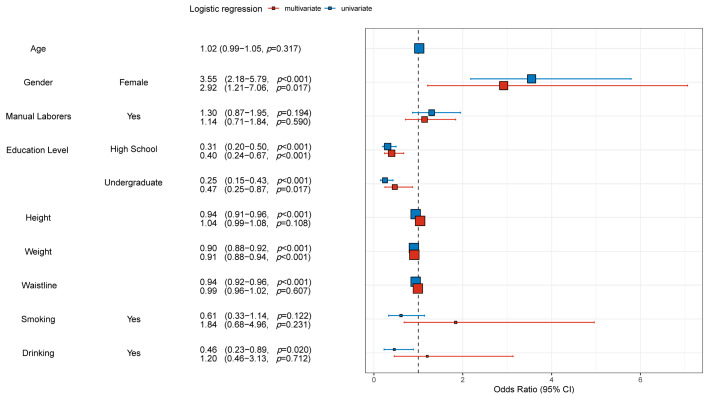
Univariate and multivariate logistic regression analyses for the training group. Univariate and multivariate logistic regression analyses were performed for all factors. Odds ratios, confidence intervals, and *p*-values were shown. *p* < 0.05 meant that the difference was statistically significant. Abbreviations: OR, Odds ratios; CI, confidence intervals.

**Figure 2 jcm-12-01292-f002:**
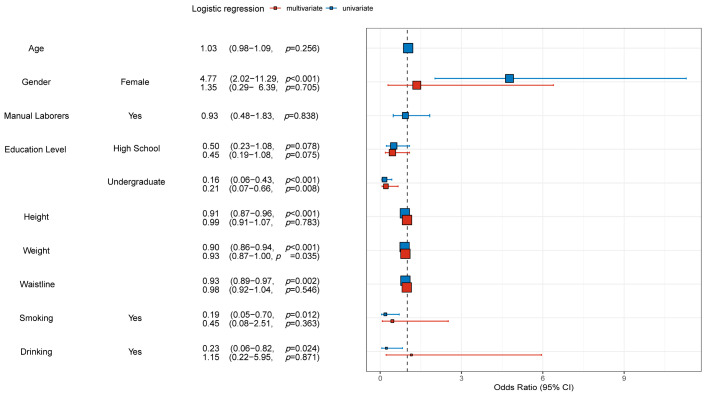
Univariate and multivariate logistic regression analyses for the validation group. Univariate and multivariate logistic regression analyses were performed for all factors. Odds ratios, confidence intervals, and *p*-values were shown. *p* < 0.05 meant that the difference was statistically significant. Abbreviations: OR, Odds ratios; CI, confidence intervals.

**Figure 3 jcm-12-01292-f003:**
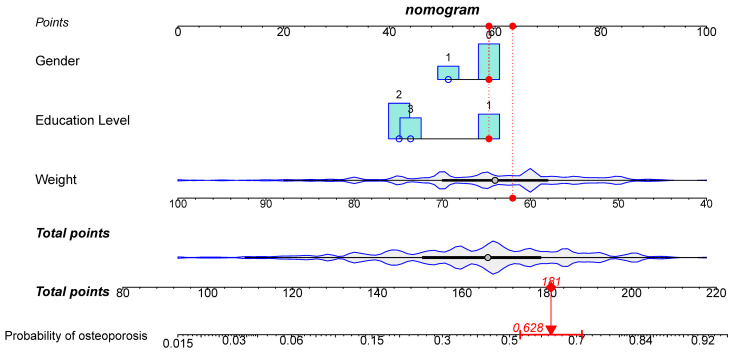
Nomogram for predicting osteoporosis based on the training group. Three clinical characteristics—gender, education level, and weight—were incorporated into the nomogram prediction model. The first row was the Point marked with a scale. Rows 2, 3, and 4 were the 3 clinical characteristics, which were marked by scales, respectively. Rows 5 and 6 were Total points marked with scales. Row 7 was the predicted probability of osteoporosis marked with a scale. The distribution of each clinical characteristic was also shown. Each clinical characteristic was marked with a red dot and a vertical line was drawn according to the specific situation of a particular clinical subject here. The vertical line intersected with the 1st line; this Point was recorded, and each Point was summed to obtain the Total point. Then, the vertical line was made at this Total point and the vertical line intersected with the 7th row. This probability here is the probability of osteoporosis of this one clinical subject. In this graph, a female clinical subject with an education level of 1 and a weight of 62 kg was used as an example. According to this nomogram clinical prediction model, the Total point of osteoporosis for this clinical subject was 181, corresponding to a probability of 0.628. The “1”, “2”, and “3” for “Education Level” stand for “Junior high school”, “High school”, and “Undergraduate”, respectively.

**Figure 4 jcm-12-01292-f004:**
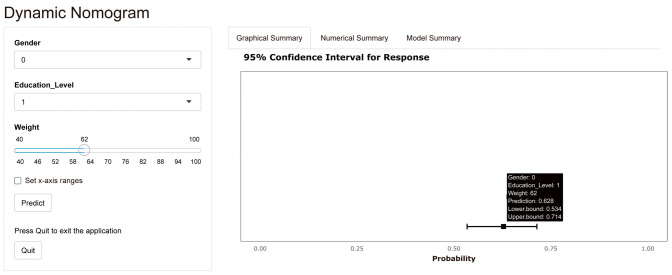
Dynamic Nomogram for predicting osteoporosis based on the training group. The Dynamic Nomogram can be used on computers and any mobile device. The specific link was: https://shibaolu.shinyapps.io/DynamicNomogram/ accessed on 22 December 2022. Here, the three specific clinical characteristics of the clinical subjects can be set via the parameter box on the left. When the “Predict” button was clicked, the predicted probability and 95% confidence interval for osteoporosis for this clinical subject can be displayed in the “Graphical Summary” box on the right. Here, a female clinical subject with an education level of 1 and a weight of 62 kg was still used as an example.” The “1”, “2”, and “3” for “Education Level” stand for “Junior high school”, “High school”, and “Undergraduate”, respectively.

**Figure 5 jcm-12-01292-f005:**
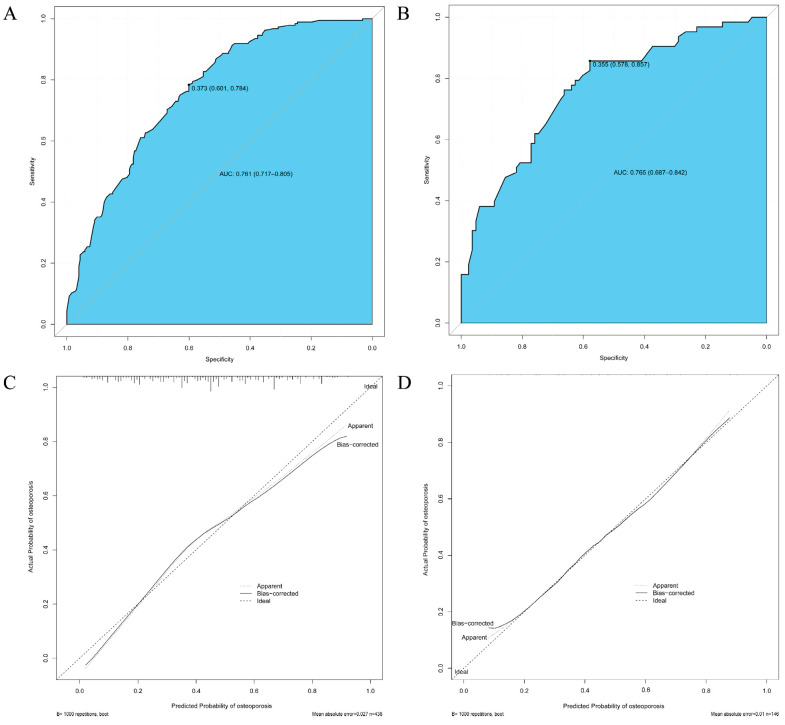
Validation of discrimination and calibration of nomogram prediction models. (**A**) ROC curve of the training group; (**B**) ROC curve of the validation group; (**A**,**B**) ROC, AUC, 95% CI, and cutoff values were shown. Higher AUC values suggested higher discrimination of the nomogram clinical prediction model. (**C**) Calibration curve of the training group; (**D**) Calibration curve of the validation group. (**C**,**D**) The horizontal coordinate was the probability of predicted osteoporosis based on the nomogram clinical prediction model. The vertical coordinate was the actual probability of osteoporosis. The Ideal line was the diagonal dashed line, representing the predicted probability equal to the actual probability under the optimal condition. The Apparent line and the Bias-corrected line, respectively, represent the predicted probability and the actual probability of the nomogram clinical prediction model developed in this study under the present real conditions. The closer these two lines were to the Ideal line (diagonal dashed line), the better the nomogram clinical prediction model was, and the better its calibration was. Abbreviations: ROC, receiver operator characteristic curve; AUC, the area under the ROC curve; CI, confidence intervals.

**Figure 6 jcm-12-01292-f006:**
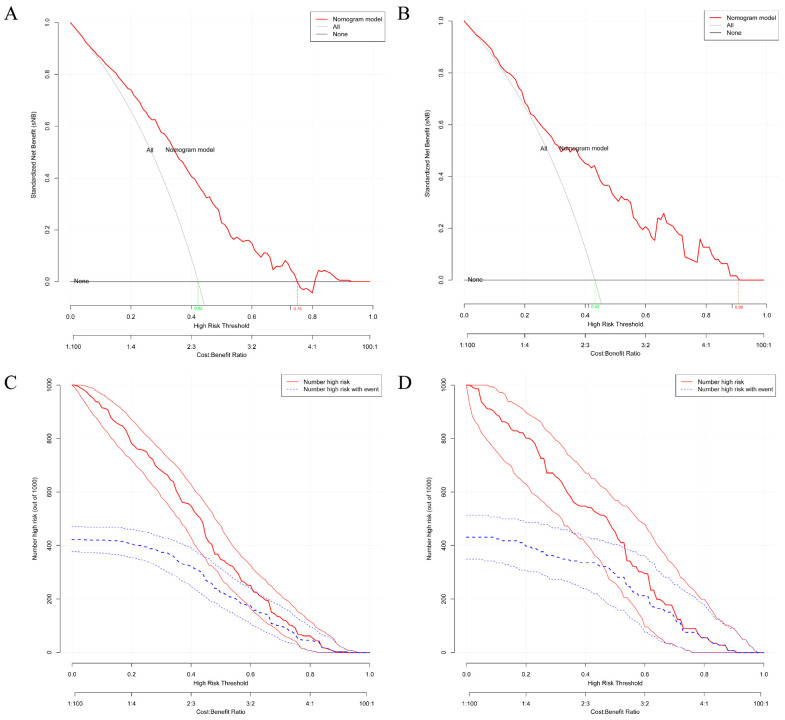
Validation of clinical benefits and clinical impacts of nomogram prediction models. (**A**) The DCA curve of the training group, the “Nomogram model” showed clinical benefit compared to “ALL” at a determination threshold of 0–0.75. (**B**) The DCA curve of the validation group, the “Nomogram model” showed clinical benefit compared to “ALL” at a determination threshold of 0–0.90. (**A**,**B**) The red line (Nomogram model) represented the overall benefit if screening was performed based on the Nomogram model clinical prediction model developed for this study. Clinical subjects with true osteoporosis who were screened (true positives) were considered to have received a clinical benefit (because early detection and diagnosis and treatment were possible after screening). Patients who did not have osteoporosis but were screened for osteoporosis (false positives) were considered to have incurred costs but did not receive clinical benefit. The trade-off between these overall costs and clinical benefits was the study objective of the DCA curve. The gray line (ALL) represented the overall benefit of considering all clinical subjects to be osteoporotic. This also yielded some clinical benefit and unnecessary cost. The gray-black line (None) represented the assumption that all clinical subjects were not osteoporotic and thus no action was taken. No cost was incurred, but there was no clinical benefit either. The horizontal coordinate represented the threshold at which clinical subjects were judged to be positive (high risk threshold, ranging from 0–1 for the “Nomogram model” and “ALL”, respectively). The vertical coordinate represented the net clinical benefit after normalization (standardized net benefit, sNB). At the same threshold, the sNB of “Nomogram model” and “ALL” were compared, and if sNB > 0, a clinical benefit was considered at this point. (**C**) Clinical impact curve of the training group. (**D**) Clinical impact curve of the validation group. (**C**,**D**) At the same horizontal coordinate (high risk threshold), the vertical coordinate (Number high risk (out of 1000)) of the red line (Number high risk) and the blue line (Number high risk with event) were compared, and if the vertical coordinate of the red line was higher than the blue line, it was considered clinically significant at that point.

**Table 1 jcm-12-01292-t001:** Baseline data and between-group (No osteoporosis and Osteoporosis) comparisons were made for the training and validation groups.

	Training Group				Validation Group			
	[All]	No Osteoporosis	Osteoporosis	*p* Value	[All]	No Osteoporosis	Osteoporosis	*p* Value
	*n =* 438	*n =* 253	*n =* 185		*n =* 146	*n =* 83	*n =* 63	
Age (years):	67.2 ± 6.48	66.9 ± 6.66	67.5 ± 6.23	0.312	66.1 ± 6.31	65.6 ± 6.94	66.8 ± 5.37	0.240
Gender:				<0.001				<0.001
Female	319 (72.8%)	160 (63.2%)	159 (85.9%)		104 (71.2%)	49 (59.0%)	55 (87.3%)	
Male	119 (27.2%)	93 (36.8%)	26 (14.1%)		42 (28.8%)	34 (41.0%)	8 (12.7%)	
Manual Laborers:				0.231				0.974
No	292 (66.7%)	175 (69.2%)	117 (63.2%)		89 (61.0%)	50 (60.2%)	39 (61.9%)	
Yes	146 (33.3%)	78 (30.8%)	68 (36.8%)		57 (39.0%)	33 (39.8%)	24 (38.1%)	
Education Level:				<0.001				0.001
Junior high school	133 (30.4%)	49 (19.4%)	84 (45.4%)		46 (31.5%)	18 (21.7%)	28 (44.4%)	
High school	192 (43.8%)	125 (49.4%)	67 (36.2%)		64 (43.8%)	36 (43.4%)	28 (44.4%)	
Undergraduate	113 (25.8%)	79 (31.2%)	34 (18.4%)		36 (24.7%)	29 (34.9%)	7 (11.1%)	
Height (cm)	160 [156;167]	162 [158;169]	160 [155;164]	<0.001	161 [157;168]	163 [160; 171]	158 [156; 162]	<0.001
Weight (kg)	64.0 [58.0; 70.0]	67.0 [60.0; 75.0]	60.0 [55.0;65.0]	<0.001	65.0 [59.0;70.0]	67.0 [61.0; 74.5]	60.0 [54.0; 65.0]	<0.001
Waistline (cm)	84.0 [79.0; 90.0]	86.0 [80.0; 92.0]	80.0 [76.0;86.0]	<0.001	84.0 [79.0; 90.0]	85.0 [80.0; 90.0]	80.0 [76.0;86.5]	0.001
Smoking:				0.160				0.013
No	388 (88.6%)	219 (86.6%)	169 (91.4%)		126 (86.3%)	66 (79.5%)	60 (95.2%)	
Yes	50 (11.4%)	34 (13.4%)	16 (8.65%)		20 (13.7%)	17 (20.5%)	3 (4.76%)	
Drinking:				0.027				0.030
No	389 (88.8%)	217 (85.8%)	172 (93.0%)		128 (87.7%)	68 (81.9%)	60 (95.2%)	
Yes	49 (11.2%)	36 (14.2%)	13 (7.03%)		18 (12.3%)	15 (18.1%)	3 (4.76%)	

Notes: Numerical variables for normality were represented as mean ± standard deviation. Non-normal numerical variables were represented by median ± upper and lower quartiles. Categorical variables were represented by frequency and percentage. *p* < 0.05 meant that the difference was statistically significant.

## Data Availability

The data can be obtained through the email under reasonable request: shibaolu@xwh.ccmu.edu.cn.

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
