# Peer review of "Construction and Validation of a Nomogram Clinical Prediction Model for Predicting Osteoporosis in an Asymptomatic Elderly Population in Beijing"

_jcm, 2023, doi:10.3390/jcm12041292_

Round 1

Reviewer 1 Report

This study aimed to construct a nomogram clinical prediction model for predicting osteoporosis in the general elderly population, in order to facilitate early detection and diagnosis of the disease.

Row 33 - "where aging is increasing 33 dramatically" - This sentence doesn't sound right - please rephrase

Row 65-66 - I believe COVID-19 "Pneumonia" should be written with "p" lowercase.

Row 122 - Osteopenia should be written with lower case "o"

Row 136 - What exactly is shinyapps? Is this something reliable in the scientific community?

As it will be the most read section of your complex analysis, I suggest emphasizing the discussion/conclusion section with the following:

- Discuss the limitations of the study and how they may have affected the results.

- Suggest directions for future research, such as examining the potential roles of other risk factors or investigating the effectiveness of different interventions in this population.

- Explore the potential clinical implications of the findings, including how the nomogram could be used in practice to identify individuals at high risk of osteoporosis.

- Discuss the potential public health implications of the study, such as the impact of identifying and treating osteoporosis in the asymptomatic elderly population on morbidity and mortality rates.

- Consider the generalizability of the study's findings to other populations, and highlight any potential differences or similarities in risk factors and outcomes that might be observed in other settings.

Author Response

Dear reviewer, thank you very much for your careful review. Your meticulous and responsible review will certainly greatly enhance the rigor and scientific quality of this manuscript. Next I will do point-to-point responses to your suggestions.

Q1: Row 33 - "where aging is increasing 33 dramatically" - This sentence doesn't sound right - please rephrase

A1: Many thanks to the rigorous reviewers, I have revised it. Specifically reflected in the manuscript: "As the population ages and the average life expectancy of people increases, the inci-dence of osteoporosis is on the rise year by year, and this is even more pronounced in China, because China is rapidly entering an aging society."

Q2: Row 65-66 - I believe COVID-19 "Pneumonia" should be written with "p" lowercase.

A2: Many thanks to the rigorous reviewers, and I have corrected this typo.

Q3: Osteopenia should be written with lower case "o"

A3: Many thanks to the rigorous reviewers, and I have corrected this typo.

Q4: What exactly is shinyapps? Is this something reliable in the scientific community?

A4: Thank you very much for the rigorous reviewers. I really should answer your question about shinyapps. "Shiny" is an open source R program that helps researchers to develop and publish online interactive applications more easily, without HTML/CSS/JavaScript. The source code link is: (https://github.com/rstudio/shiny#readme). After developing an interactive program in R, the researcher can publish it at https://www.shinyapps.io/ and then generate a link to a specific URL that can be made public, such as the one published in this manuscript (https://shibaolu.shinyapps.io/DynamicNomogram/.). As you can see, "shibaolu" is my account, so the link to the application generated through my account is “shibaolu.shinyapps.io”. And "DynamicNomogram" is a name I use for the online interactive application.

A search of Pubmed using the keyword “shinyapps” yielded 560 documents, with an increasing trend year by year. There are many valuable online interactive programs built on shinyapps, including some online interactive nomogram programs (A search in Pubmed using the keywords "(shinyapps) AND (nomogram)" even yielded 126 related studies.). One of the more famous is the TIMER database (Tumor Immune Estimation Resource, cistrome.shinyapps.io/timer). This database is an online interactive program based on shinyapps and was released in 2017. Its developer's interpretation of the database is published in the relevant literature (PMID:29092952; PMCID:PMC6042652; DOI:10.1158/0008-5472.CAN-17-0307). This TIMER database enables researchers to comprehensively explore the molecular features of tumor immune interactions. A search of Pubmed with the keyword "Tumor Immune Estimation Resource" has retrieved 765 papers. This is a good indication of the scientific nature of the TIMER database and indirectly demonstrates the great benefit of shinyapps for the development of online interactive programs.

Looking back now, although our online interactive program is a little cruder than this TIMER database, it still has its scientific nature to a certain extent.

Q5- Q9: Thank you very much to the distinguished reviewers. We have revised the discussion section as you suggested to make it more insightful.

Q5: Discuss the limitations of the study and how they may have affected the results.

A5: Many thanks to the rigorous reviewers. We discuss limitations of the study and give examples. Relevant revisions are specified at the end of the fourth paragraph of the “Discussion” section. Specifically reflected in the manuscript as:“ On the other hand, there may be more potential risk factors to consider in other regions or populations. Because if a more comprehensive risk factor is included, it will inevitably affect the validation indicators such as the discrimination and calibration of the nomogram model. This is also a great challenge to the generalizability of the nomogram clinical prediction model developed in this study. For example, in areas with unbalanced economic development, the economic level of the subjects should be considered. Other examples may include the subject's exercise status, nutritional intake and use of anti-osteoporosis drugs. Therefore, it is urgent to explore the potential risk factors of osteoporosis more comprehensively in subsequent studies using populations from other regions as validation groups, so as to more reasonably validate the nomogram clinical prediction model. In addition, the research outcome of this study is the occurrence of osteoporosis, but to some extent, the three-year, five-year or ten-year incidence rate of osteoporosis can be explored by exploring the results with a time span. It is also a very meaningful research direction in the future.”

Q6: Suggest directions for future research, such as examining the potential roles of other risk factors or investigating the effectiveness of different interventions in this population.

A6: Many thanks to the rigorous reviewers. We discuss directions for future research and give examples. Relevant revisions are specified at the end of the fourth paragraph of the “Discussion” section. Specifically reflected in the manuscript as:“ On the other hand, there may be more potential risk factors to consider in other regions or populations. Because if a more comprehensive risk factor is included, it will inevitably affect the validation indicators such as the discrimination and calibration of the nomogram model. This is also a great challenge to the generalizability of the nomogram clinical prediction model developed in this study. For example, in areas with unbalanced economic development, the economic level of the subjects should be considered. Other examples may include the subject's exercise status, nutritional intake and use of anti-osteoporosis drugs. Therefore, it is urgent to explore the potential risk factors of osteoporosis more comprehensively in subsequent studies using populations from other regions as validation groups, so as to more reasonably validate the nomogram clinical prediction model. In addition, the research outcome of this study is the occurrence of osteoporosis, but to some extent, the three-year, five-year or ten-year incidence rate of osteoporosis can be explored by exploring the results with a time span. It is also a very meaningful research direction in the future.”

Q7: Explore the potential clinical implications of the findings, including how the nomogram could be used in practice to identify individuals at high risk of osteoporosis.

A7: Many thanks to the rigorous reviewers. We discuss the clinical significance of the nomogram clinical prediction model and how to use it in practice, and give examples. Relevant revisions are specified at the end of the third paragraph of the “Discussion” section. Specifically reflected in the manuscript as :“Through the common and online dynamic interactive nomogram prediction model constructed in this study, a quick and simple screening of asymptomatic elderly people in the community could be carried out in practice. If the subject's "probability of osteoporosis" is high (0.74), it should be recommended that the subject undergo DXA ex-amination as much as possible, which can improve the detection rate of osteoporosis to a certain extent. If the subject is indeed in the early stage of osteoporosis, for example, the T-score of BMD in the later test is -2.6 SD, the subject could be given certain anti-osteoporosis treatment, so as to achieve early detection of osteoporosis and early treatment, thereby reducing the rate of osteoporotic fracture and mortality. And if the subject's "probability of osteoporosis" is medium (0.51), and the subject has a high probability of developing osteoporosis, for example, the T-score of BMD in the subsequent test is -2.3 SD, the subject could be given certain treatment to prevent osteoporosis, thereby reducing the incidence of osteoporosis. The above discussion about the potential impact of this study on osteoporosis morbidity and mortality in practice demonstrates its potential clinical significance and value to public health.”

Q8: Discuss the potential public health implications of the study, such as the impact of identifying and treating osteoporosis in the asymptomatic elderly population on morbidity and mortality rates.

A8: Many thanks to the rigorous reviewers. We discuss and give examples of the impact of nomogram clinical prediction models on public health (morbidity and mortality). Relevant revisions are specified at the end of the third paragraph of the “Discussion” section. Specifically reflected in the manuscript as:“ Through the common and online dynamic interactive nomogram prediction model constructed in this study, a quick and simple screening of asymptomatic elderly people in the community could be carried out in practice. If the subject's "probability of osteoporosis" is high (0.74), it should be recommended that the subject undergo DXA ex-amination as much as possible, which can improve the detection rate of osteoporosis to a certain extent. If the subject is indeed in the early stage of osteoporosis, for example, the T-score of BMD in the later test is -2.6 SD, the subject could be given certain anti-osteoporosis treatment, so as to achieve early detection of osteoporosis and early treatment, thereby reducing the rate of osteoporotic fracture and mortality. And if the subject's "probability of osteoporosis" is medium (0.51), and the subject has a high probability of developing osteoporosis, for example, the T-score of BMD in the subsequent test is -2.3 SD, the subject could be given certain treatment to prevent osteoporosis, thereby reducing the incidence of osteoporosis. The above discussion about the potential impact of this study on osteoporosis morbidity and mortality in practice demonstrates its potential clinical significance and value to public health.”

Q9: Consider the generalizability of the study's findings to other populations, and highlight any potential differences or similarities in risk factors and outcomes that might be observed in other settings.

A9: Many thanks to the rigorous reviewers. We discuss the generalizability of the findings and give examples. Relevant revisions are specified at the end of the fourth paragraph of the “Discussion” section. Specifically reflected in the manuscript as:“ On the other hand, there may be more potential risk factors to consider in other regions or populations. Because if a more comprehensive risk factor is included, it will inevitably affect the validation indicators such as the discrimination and calibration of the nomogram model. This is also a great challenge to the generalizability of the nomogram clinical prediction model developed in this study. For example, in areas with unbalanced economic development, the economic level of the subjects should be considered. Other examples may include the subject's exercise status, nutritional intake and use of anti-osteoporosis drugs. Therefore, it is urgent to explore the potential risk factors of osteoporosis more comprehensively in subsequent studies using populations from other regions as validation groups, so as to more reasonably validate the nomogram clinical prediction model. In addition, the research outcome of this study is the occurrence of osteoporosis, but to some extent, the three-year, five-year or ten-year incidence rate of osteoporosis can be explored by exploring the results with a time span. It is also a very meaningful research direction in the future.”

Reviewer 2 Report

The article is well written and the topic is very interesting. I really appreciate the R codes as supplementary file.

I suggest only few elements:

-          Pag. 3, line 105. The Authors should clarified the variable “history of drinking” (amount, frequency…of alcohol consumption)

-          Pag 3, line 109-1010 the measurements were performed only one time or repeated?

-          Pag 3, line 124. Please change osteopenia with osteoporosis

-          Pag. 4, line 160. The Authors should provide more details regarding continuous variables transformation

-          I really appreciate the model and its performance evaluation along with the validation with the use of a control group. However, it could be more useful to predict the osteoporosis occurrence (years) instead of confirming its presence. The authors should provide more information about this topic in the discussion.

Author Response

Dear reviewer, thank you very much for your careful review. Your meticulous and responsible review will certainly greatly enhance the rigor and scientific quality of this manuscript. Next I will do point-to-point responses to your suggestions.

Q1: Pag. 3, line 105. The Authors should clarified the variable “history of drinking” (amount, frequency…of alcohol consumption)

A1: Many thanks to the rigorous reviewers. We have made detailed clarifications and revisions to "history of drinking". Specifically reflected in the manuscript: "History of drinking was defined as: drinking liquor at least once a week, more than 50ml each time, lasting more than one year, still drinking or not quitting alcohol for more than 3 years."

Q2: Pag 3, line 109-1010 the measurements were performed only one time or repeated?

A2: Many thanks to the rigorous reviewers. We clarify and revise it in detail. Specifically reflected in the manuscript: "The relevant physical measurements were repeated twice and the results were averaged."

Q3: Pag 3, line 124. Please change osteopenia with osteoporosis

A3: Many thanks to the rigorous reviewers, and I have corrected this typo.

Q4: Pag. 4, line 160. The Authors should provide more details regarding continuous variables transformation

A4: Many thanks to the rigorous reviewers. We clarify and revise it in detail. Specifically reflected in the manuscript: "Continuous variables were transformed into dichotomous variables according to their means, and then analyzed for interaction effects."

Q5: However, it could be more useful to predict the osteoporosis occurrence (years) instead of confirming its presence. The authors should provide more information about this topic in the discussion.

A5: Many thanks to the rigorous reviewers. We discuss the value of another study of osteoporosis with a time span. Relevant revisions are specified at the end of the fourth paragraph of the “Discussion” section. Specifically reflected in the manuscript as:“ In addition, the research outcome of this study is the occurrence of osteoporosis, but to some extent, the three-year, five-year or ten-year incidence rate of osteoporosis can be explored by exploring the results with a time span. It is also a very meaningful research direction in the future.”
